# A Simple Frequency Formulation for the Tangent Oscillator

**Ji-Huan He** [1,2,3,*], **Qian Yang** [1], **Chun-Hui He** [4] and **Yasir Khan** [5]

1   School of Science, Xi'an University of Architecture and Technology, Xi'an 710055, China; yangq@xauat.edu.cn
2   School of Mathematics and Information Science, Henan Polytechnic University, Jiaozuo 454150, China
3   National Engineering Laboratory for Modern Silk, College of Textile and Clothing Engineering, Soochow University, 199 Ren-Ai Road, Suzhou 215123, China
4   School of Civil Engineering, Xi'an University of Architecture and Technology, Xi'an 710055, China; Mathew_He@yahoo.com
5   Department of Mathematics, University of Hafr Al-Batin, Hafr Al-Batin 31991, Saudi Arabia; yasirmath@yahoo.com
*   Correspondence: hejihuan@suda.edu.cn

**Abstract:** The frequency of a nonlinear vibration system is nonlinearly related to its amplitude, and this relationship is critical in the design of a packaging system and a microelectromechanical system (MEMS). This paper proposes a straightforward frequency prediction method for nonlinear oscillators with arbitrary initial conditions. The tangent oscillator, the hyperbolic tangent oscillator, a singular oscillator, and a MEMS oscillator are chosen to elucidate the simple solving process. The results, when compared with those obtained by the homotopy perturbation method, exhibit a good agreement. This paper introduces a very convenient procedure for attaining quick and accurate insight into the vibration property of a nonlinear vibration system.

**Keywords:** nonlinear vibration; period-amplitude relationship; He's frequency formulation; tangent oscillator; homotopy perturbation method; microelectromechanical system (MEMS)

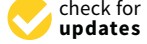



## 1. Introduction

Vibration absorption and vibration attenuation are two critical factors in designing a nonlinear vibration system; for example, a low amplitude is always considered in the design of the packaging system [1–4] and the seismic design of architecture [5]. Active vibration control [6] has received a lot of attention in the industrial and academic communities, and many mathematicians are working to predict the periodic property of a practical vibration system. Generally, a nonlinear vibration equation is written as

$$mw'' + h(w) = 0, \; w(0) = a, w'(0) = b \tag{1}$$

where $w$ is the displacement, $m$ is the mass, $h$ is the nonlinear restoring force, and $a$ and $b$ are constants. For a linear vibration system, one can choose $h(w) = kw$, $k$ is the spring coefficient, this is the well-known harmonic oscillation. When $h(w) = k \tan w$, we have the well-known tangent oscillator arising in packaging systems [1–4]. The amplitude is determined by the initial conditions and the frequency of the system. There are numerous analytical methods for solving Equation (1), such as the Li–He method or its modifications [4,7,8], He's variational approach [9], He's Energy Balance Method [10], the iteration perturbation method [11], the exact solution method [12], the homotopy perturbation method (HMP) [13,14], the Gamma function method [15], and the variational method [16–18]. The goal of this paper is to suggest a simple method [19,20] for gaining a timely and efficient glimpse into the frequency–amplitude relationship of Equation (1) with arbitrary initial conditions. The comparison with the aforementioned existing methods shows that this work would be greatly challenging for nonlinear vibration theory.

## 2. The Frequency Formulation

In Equation (1), $h(w)/w > 0$ is required for a periodic solution. Equation (1) can be rewritten as follows:

$$w'' + h(w) = 0, \ w(0) = a, w'(0) = b \tag{2}$$

In terms of frequency formulation [19,20] is

$$\omega^2 = \left. \frac{h(w)}{w} \right|_{w = \pm NA} \tag{3}$$

where $A$ is the amplitude, it can be approximated calculated as

$$A = \sqrt{a^2 + \frac{b^2}{\omega^2}} \tag{4}$$

In Equation (3) $N$ was recommended as $\sqrt{3}/2$ in [19] for non-singular oscillator and 0.8 in [20] for singular oscillators. The frequency can be calculated using Equations (3) and (4). In the literature [21–23], Equation (3) is referred to as He's frequency formulation, and there have been numerous modifications [24–27]; the original formulation began with an ancient Chinese algorithm [28], which led to He's frequency formulation [19,20] for nonlinear vibration systems and Chun-Hui He's algorithm [28,29] in numerical methods.

## 3. Tangent Oscillator

The tangent oscillator [3] is

$$w'' + \omega_0^2 \tan w = 0, \ w(0) = a, w'(0) = b \tag{5}$$

One can choose $N = \sqrt{3}/2$ in Equation (3). We have

$$\omega^2 = \frac{\omega_0^2 \tan(\sqrt{3}A/2)}{\sqrt{3}A/2} \tag{6}$$

where $A$ is given in Equation (4). For given parameters of $\omega_0$, $a$, and $b$, the frequency can be calculated easily from Equation (6). To obtain an explicit formulation, one can consider the case of $A << 1$. In that case, we have

$$\tan w \approx w + \frac{1}{3}w^3 \tag{7}$$

Equation (6) becomes

$$\omega^2 = \frac{\omega_0^2 \tan(\sqrt{3}A/2)}{\sqrt{3}A/2} = \omega_0^2(1 + \frac{1}{4}A^2) \tag{8}$$

In view of Equation (4), we can convert Equation (8) to the form

$$\omega^2 = \omega_0^2 \left[ 1 + \frac{1}{4}(a^2 + \frac{b^2}{\omega^2}) \right] \tag{9}$$

or

$$4\omega^4 - (4 + a^2)\omega_0^2\omega^2 - b^2\omega_0^2 = 0 \tag{10}$$

Solving $\omega$ from Equation (10) and ignoring the meaningless root gives

$$\omega = \sqrt{\frac{(4 + a^2)\omega_0^2 + \sqrt{(4 + a^2)^2\omega_0^4 + 16b^2\omega_0^2}}{8}} \tag{11}$$

The approximate solution reads

$$w = A\cos(\omega t + \varphi) \tag{12}$$

where $\varphi$ is determined by the initial conditions

$$\varphi = \arctan(-\frac{b}{a\omega}) \tag{13}$$

In order to reveal the accuracy of the approximate solution, we resolve the problem by the homotopy perturbation method [19]. For small amplitude, Equation (5) is approximated as

$$w'' + \omega_0^2(w + \frac{1}{3}w^3) = 0, \ w(0) = a, w'(0) = b \tag{14}$$

The homotopy equation is

$$x'' + \omega^2 x + h\left[\omega_0^2 x - \omega^2 x + \frac{1}{3}\omega_0^2 x^3\right] = 0 \tag{15}$$

where $h$ is the homotopy parameter, $0 \leq h \leq 1$. When $h$ = 1, Equation (15) becomes Equation (14). We assume that the solution is written as

$$w = w_0 + hw_1 + h^2 w_2 + \cdots \tag{16}$$

Following the method's standard steps, we have

$$w''_0 + \omega^2 w_0 = 0, w_0(0) = a, w'_0(0) = b \tag{17}$$

$$w''_{11} + \omega^2 w_1 + \omega_0^2 w_0 - \omega^2 w_0 + \frac{1}{3}\omega_0^2 w_0^3 = 0, w_1(0) = 0, w'_0(0) = 0 \tag{18}$$

Equation (17) is a linear differential equation; its solution is

$$w_0 = A\cos(\omega t + \varphi) \tag{19}$$

where $A$ and $\varphi$ are given, respectively, in Equations (4) and (13). Now Equation (18) is updated as

$$w''_1 + \omega^2 w_1 + (\omega_0^2 - \omega^2)A\cos(\omega t + \varphi) + \frac{1}{3}\omega_0^2 A^3 \cos^3(\omega t + \varphi) = 0 \tag{20}$$

or

$$w''_1 + \omega^2 w_1 + (\omega_0^2 - \omega^2 + \frac{1}{4}\omega_0^2 A^2)A\cos(\omega t + \varphi) + \frac{1}{12}\omega_0^2 A^3 \cos(3\omega t + 3\varphi) = 0 \tag{21}$$

The solution of Equation (21) is

$$w_1 = -\frac{1}{2\omega}(\omega_0^2 - \omega^2 + \frac{1}{4}\omega_0^2 A^2)At\sin(\omega t + \varphi) + \frac{1}{96\omega^2}\omega_0^2 A^3 \cos(3\omega t + 3\varphi) + C \tag{22}$$

where $C$ is an integral constant. In Equation (22), the term of $t\sin(\omega t + \varphi)$ is not periodic when $t$ tends to infinity, so its coefficient has to be zero to have a periodic solution, that is

$$\omega_0^2 - \omega^2 + \frac{1}{4}\omega_0^2 A^2 = 0 \tag{23}$$

or

$$\omega = \omega_0 \sqrt{1 + \frac{1}{4}A^2} \tag{24}$$

This result can also be obtained by the multiple scales method [24], and it is exactly the same as that given in Equation (9). Considering the initial conditions, $w_1(0) = 0, w'_0(0) = 0$, Equation (22) becomes

$$w_1 = \frac{1}{96\omega^2}\omega_0^2 A^3 [\cos(3\omega t + 3\varphi) - \cos(\omega t + \varphi)] \tag{25}$$

The first-order approximate solution is obtained by setting $h = 1$ in Equation (16), that is

$$w = w_0 + w_1 = A\cos(\omega t + \varphi) + \frac{1}{96\omega^2}\omega_0^2 A^3 [\cos(3\omega t + 3\varphi) - \cos(\omega t + \varphi)] \tag{26}$$

Comparison of the approximate solution, $w = A\cos(\omega t + \varphi)$, with $\omega$ given in Equation (11), with the exact solution for various cases is illustrated in Figure 1, where $\omega_0 = 1$.

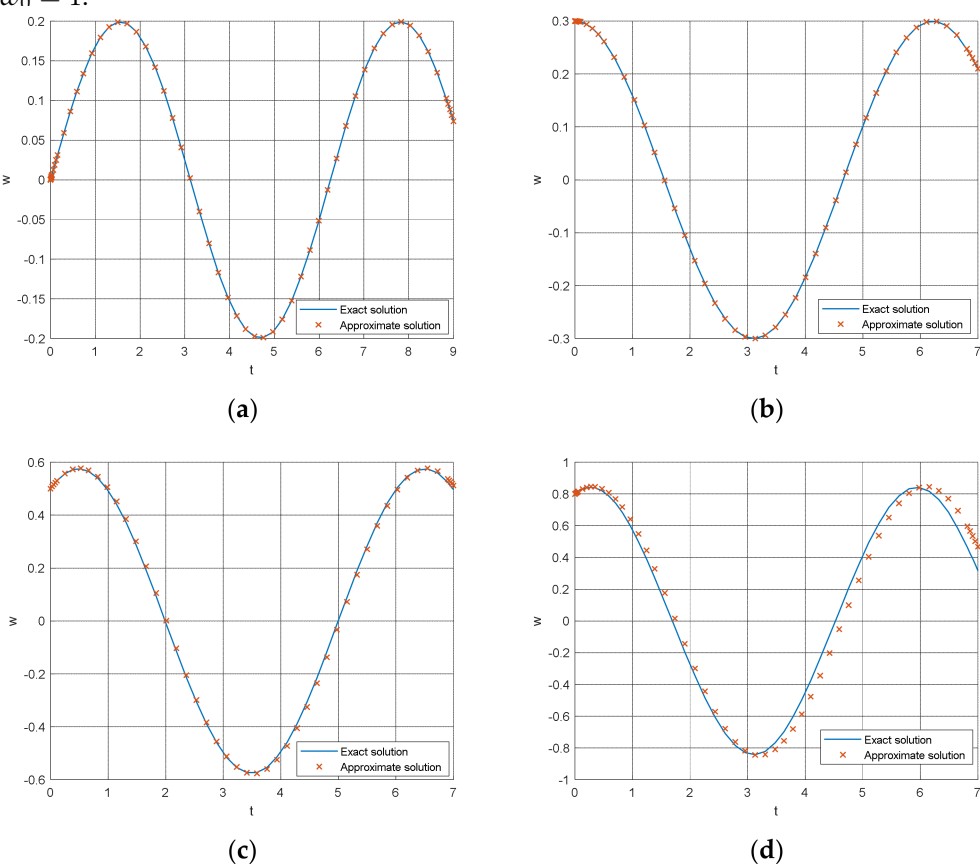

**Figure 1.** The approximate solution of Equation (12) vs. the exact one of Equation (5). (**a**) $(a,b) = (0, 0.2)$; (**b**) $(a,b) = (0.3, 0)$; (**c**) $(a,b) = (0.5, 0.3)$; and (**d**) $(a,b) = (0.8, 0.3)$.

## 4. Hyperbolic Tangent Oscillator

The hyperbolic tangent oscillator reads [1]

$$w'' + \omega_0^2 \tanh w = 0, \ w(0) = a, w'(0) = b \tag{27}$$

We can immediately obtain the following frequency–amplitude relation:

$$\omega^2 = \frac{\omega_0^2 \tanh(\sqrt{3}A/2)}{\sqrt{3}A/2} \tag{28}$$

where $A$ is defined in Equation (4).

For small amplitude, $\tanh w$ can be approximated as

$$\tanh w \approx w - \frac{1}{3}w^3 \tag{29}$$

Equation (28) becomes:

$$\omega^2 = \frac{\omega_0^2 \tanh(\sqrt{3}A/2)}{\sqrt{3}A/2} = \omega_0^2(1 - \frac{1}{4}A^2) \tag{30}$$

In view of Equation (4), Equation (30) turns out to be

$$\omega^2 = \omega_0^2\left[1 - \frac{1}{4}(a^2 + \frac{b^2}{\omega^2})\right] \tag{31}$$

or

$$4\omega^4 - (4 - a^2)\omega_0^2\omega^2 + b^2\omega_0^2 = 0 \tag{32}$$

Solving $\omega$ from Equation (32) and ignoring the meaningless root gives

$$\omega = \sqrt{\frac{(4 - a^2)\omega_0^2 + \sqrt{(4 - a^2)^2\omega_0^4 - 16b^2\omega_0^2}}{8}} \tag{33}$$

This is the same as that obtained by the homotopy perturbation method [1]. Comparison of the approximate solution with the exact solution for various cases is illustrated in Figure 2, where $\omega_0 = 1$.

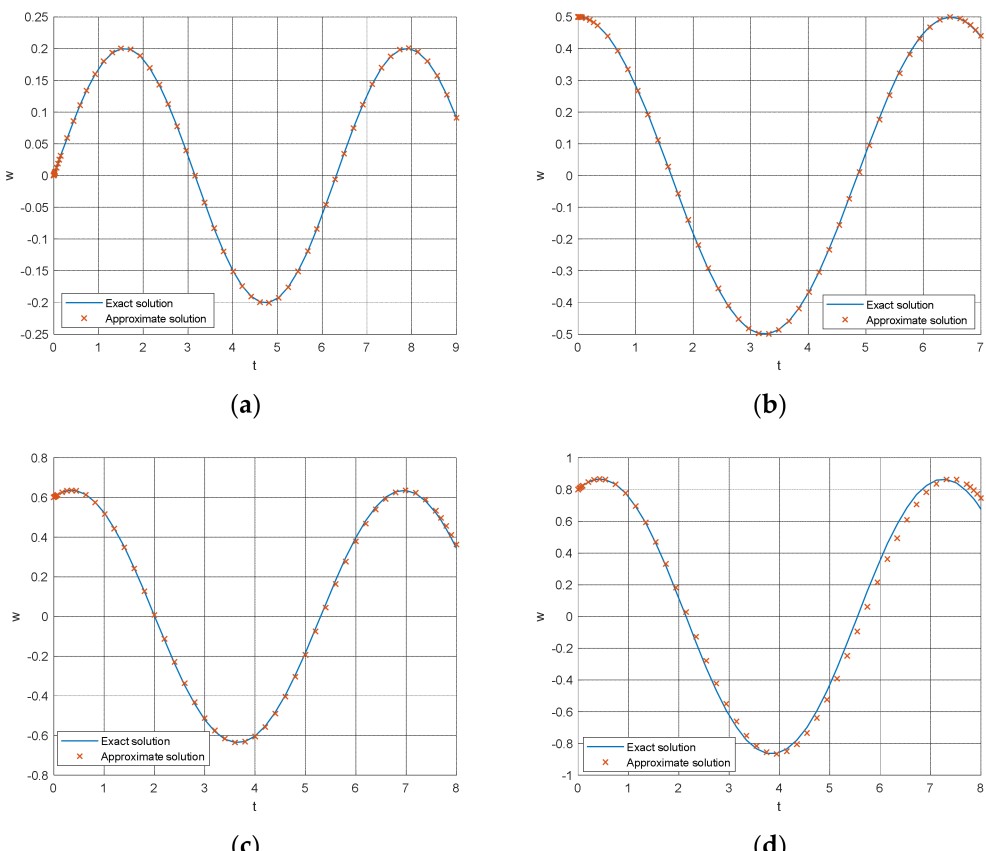

(a)

(b)

(c)

(d)

**Figure 2.** The approximate solution ($w = A\cos(\omega t + \varphi)$) vs. the exact one of Equation (27). (**a**) $(a,b) = (0, 0.2)$; (**b**) $(a,b) = (0.5, 0)$; (**c**) $(a,b) = (0.6, 0.2)$; and (**d**) $(a,b) = (0.8, 0.3)$.

## 5. Singular Oscillator

Now we consider a singular oscillator [9]

$$w'' + \frac{1}{kw} = 0, \ w(0) = a, w'(0) = b \tag{34}$$

For the singular oscillator, we choose $N = 0.8$ [20], that is

$$\omega^2 = \left.\frac{1}{kw^2}\right|_{w = NA} = \frac{1}{kN^2A^2} = \frac{1}{kN^2\left(a^2 + \frac{b^2}{\omega^2}\right)} \tag{35}$$

Simplifying Equation (35) gives

$$kN^2(a^2\omega^2 + b^2) = 1 \tag{36}$$

Solving $\omega$ from Equation (36) leads to the result

$$\omega = \frac{1}{a}\sqrt{\frac{1}{kN^2} - b^2} \tag{37}$$

When $b = 0$ and $N = 0.8$, we have

$$\omega = \frac{1}{a}\sqrt{\frac{1}{kN^2}} = 1.25k^{-1/2}a^{-1} \tag{38}$$

The exact frequency for $b = 0$ is [9]

$$\omega_{\text{exact}} = 1.2533k^{-1/2}a^{-1} \tag{39}$$

The relative error is 0.26%. Figure 3 illustrates the accuracy of the approximate solution.

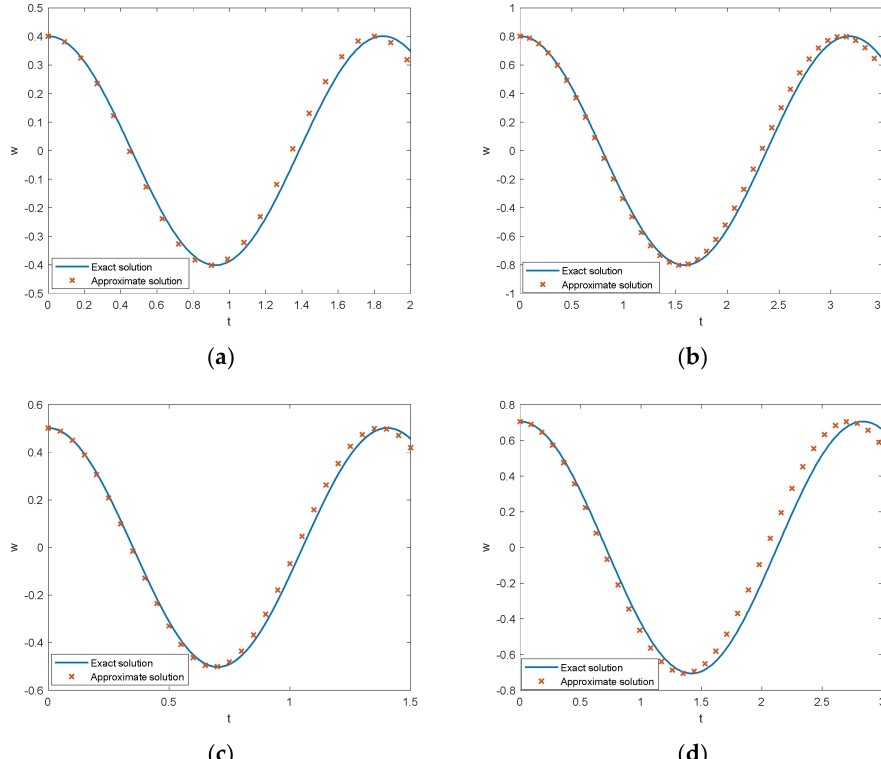

(a)

(b)

(c)

(d)

**Figure 3.** *Cont.*

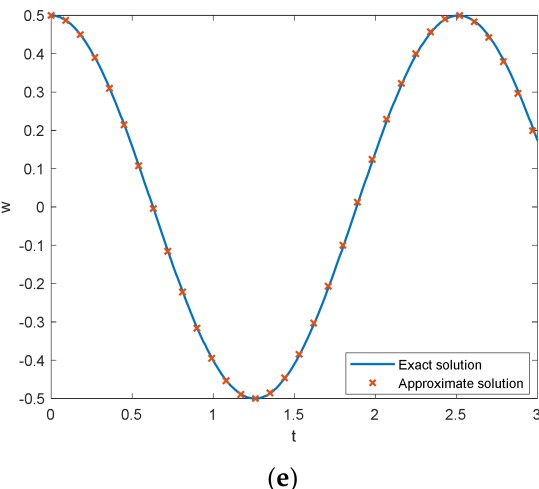

(**e**)

**Figure 3.** The comparison of approximate solution $w = A\cos(\omega t + \varphi)$ with exact one of Equation (34). (**a**) $(k,a,b) = (0.8, 0.4, 0.1)$; (**b**) $(k,a,b) = (0.6, 0.8, 0.1)$; (**c**) $(k,a,b) = (0.3, 0.5, 0.2)$; (**d**) $(k,a,b) = (0.6, 0.7, 0.2)$; and (**e**) $(k,a,b) = (1, 0.5, 0)$.

## 6. MEMS Oscillator

The fast development of nanotechnology and material science have led to skyrocketing interest in MEMS systems for the last decade [30–32]. We consider the following MEMS oscillator [33]

$$y'' + y - \frac{b}{1 - y} = 0, y(0) = 0, y'(0) = 0 \tag{40}$$

where $y$ is the dimensionless displacement, $y < 1$, and b is constant.

In order to use the above frequency formulation, we introduce a transformation:

$$y = A - x \tag{41}$$

where $A$ is the amplitude, Equation (40) becomes

$$x'' + x + \frac{b}{1 - A + x} - A = 0 \tag{42}$$

where $f(x)$ is the restoring force,

$$f(x) = x + \frac{b}{1 - A + x} - A \tag{43}$$

It requires $f(0) = 0$, that is

$$\frac{b}{1 - A} = A \tag{44}$$

or

$$A = \frac{1 - \sqrt{1 - 4b}}{2} \tag{45}$$

In view of Equation (44), we can rewrite Equation (43) in the form

$$f(x) = x - \frac{bx}{(1 - A)(1 - A + x)} \tag{46}$$

$$\omega^2 = \left. \frac{f(x)}{x} \right|_{x = -0.8A} \tag{47}$$

This leads to the following formulation

$$\omega^2 = 1 - \frac{b}{(1-A)(1-A-0.8A)} \tag{48}$$

Solving $\omega$ from Equation (48), we obtain

$$\omega = \sqrt{\frac{1 - 5.6b + \sqrt{1-4b}}{1 - 3.6b + \sqrt{1-4b}}} \tag{49}$$

Finally, we obtain the following approximate periodic solution:

$$x(t) = A\cos(\omega t) \tag{50}$$

By the inverse transformation of Equation (41), we obtain

$$y(t) = A(1 - \cos(\omega t)) = 2A\sin^2\left(\frac{\omega}{2}t\right) \tag{51}$$

where $\omega$ is given in Equation (49). Figure 4 shows a high accuracy of Equation (51) when $b < 0.15$.

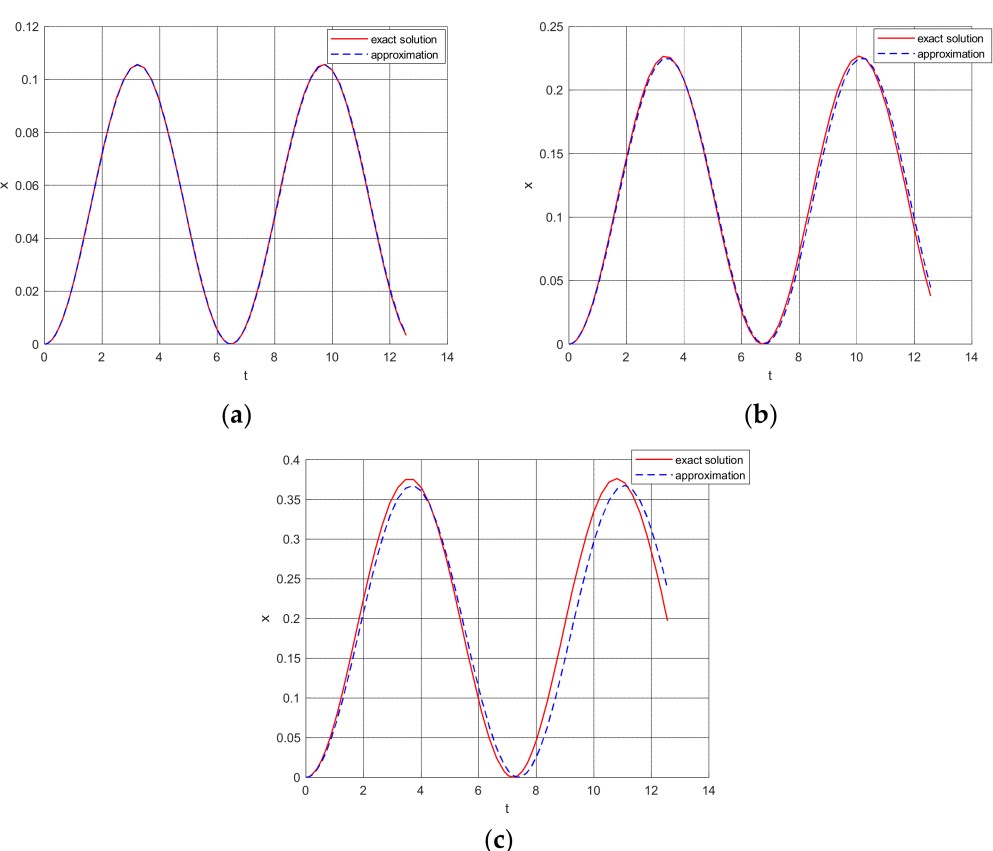

**Figure 4.** The MEMS oscillator with different values of $b$. (**a**) $b = 0.05$; (**b**) $b = 0.10$; and (**c**) $b = 0.15$.

## 7. Conclusions

He's frequency formulation could be extended to fractal oscillators [34] as well as non-conservative oscillators [35–37]. In engineering, a simple calculation is always appreciated; the simpler the calculation, the better. This paper proposes possibly the simplest method for quickly inspecting the frequency property of a nonlinear oscillator; the one-step solution yields a highly accurate result, which is quite palatable.

**Author Contributions:** Conceptualization, J.-H.H. and C.-H.H.; methodology, Q.Y. and C.-H.H.; software, Q.Y. and Y.K.; validation, J.-H.H. and C.-H.H.; formal analysis, J.-H.H.; investigation, J.-H.H.; resources, J.-H.H.; data curation, J.-H.H.; writing—original draft preparation, Q.Y. and C.-H.H.; writing—review and editing, J.-H.H. and Y.K.; visualization, J.-H.H.; supervision, J.-H.H.; project administration, J.-H.H.; funding acquisition, J.-H.H. All authors have read and agreed to the published version of the manuscript.

**Funding:** This research received no external funding.

**Institutional Review Board Statement:** Not applicable.

**Informed Consent Statement:** Not applicable.

**Data Availability Statement:** Not applicable.

**Acknowledgments:** This work was supported by Taif University Researches Supporting Project number (TURSP-2020/326), Taif University, Taif, Saudi Arabia.

**Conflicts of Interest:** The authors declare no conflict of interest.

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
