# Peer review of "A Simple Frequency Formulation for the Tangent Oscillator"

_axioms, doi:10.3390/axioms10040320_

Round 1

Reviewer 1 Report

In this paper, the authors proposed a simple frequency prediction for nonlinear oscillators with arbitrary initial conditions. They studied the tangent oscillator, the hyperbolic tangent oscillator, a singular oscillator, and the MEMS oscillator. The results are compared with those obtained from the homotopy perturbation method.

Since the frequency of a nonlinear vibrating system is nonlinearly related to its amplitude, and their relationship is critical in designing a packaging system, the obtained results could have interesting practical applications.

The paper needs a major revision.

1). Verify and correct Eq. (11). Instead of (-16b0^2omega_0^2 ) must be (+16b0^2omega_o^2). All results that are linked with Eq. (11) must be re-verified because they could be affected.

2). The abbreviation "MEMS" should be explained. Not all readers are experts in oscillating systems.

3). From (43)-(45) cannot obtain (46). Verify and correct results from this paragraph.

Author Response

In this paper, the authors proposed a simple frequency prediction for nonlinear oscillators with arbitrary initial conditions. They studied the tangent oscillator, the hyperbolic tangent oscillator, a singular oscillator, and the MEMS oscillator. The results are compared with those obtained from the homotopy perturbation method.

Since the frequency of a nonlinear vibrating system is nonlinearly related to its amplitude, and their relationship is critical in designing a packaging system, the obtained results could have interesting practical applications.

Reply: Thanks for the positive comment.

The paper needs a major revision.

1). Verify and correct Eq. (11). Instead of (-16b0^2omega_0^2 ) must be (+16b0^2omega_o^2). All results that are linked with Eq. (11) must be re-verified because they could be affected.

Reply: Thanks for the careful reading. The equation is carefully checked and corrected.

2). The abbreviation "MEMS" should be explained. Not all readers are experts in oscillating systems.

Reply: Thanks for the suggestion. Done as requested.

3). From (43)-(45) cannot obtain (46). Verify and correct results from this paragraph.

Reply: Thanks for the careful reading,  the solving process is carefully checked and corrected.

Reviewer 2 Report

The paper needs minor revision. Please see the attached file.

Author Response

Commenta on the paper
“A simple frequency formulation for the tangent oscillator”
by Ji-Huan He, Qian Yang, Chun-Hui He and Khaled A. Gepreel
The authors have proposed a simple method for analysis of nonlinear oscillator.
The paper needs minor revision.
1. The abbreviation MEMS should be explained in the paper.

Reply: Thanks for the suggestion. Done as requested. 

2. A list of References should be prepared according to the MDPI rules (the use of capital 
letters, the use of journal titles abbreviations, etc.

Reply: Thanks for the suggestion . Done as requested

Reviewer 3 Report

A research article (manuscript ID: axioms-1454856) entitled “A simple frequency formulation for the tangent oscillator” by a collaborative team from China and Saudi Arabia was submitted to the MDPI Axioms Journal.

This submitted paper has 15 pages including 4 figure and 37 references. In their report, the authors suggest a simple frequency prediction for nonlinear oscillators with arbitrary initial conditions.

The reviewer has looked through the paper and found that this presentation requires some revision.

First of all, the author must significantly improve the English language of their paper.

Second, the References must be in the MDPI format.

Also, the Graphical Abstract should be deleted because this figure represents Figure 1a.

Figures 1, 2, 3, and 4 have 4, 4, 5, and 3 figures, respectively. They should be placed in two columns instead of the single column. Also, each figure of four should be situated on the same page.

All the figure titles must start with “the”, for instance:

Figure 3. The comparison…

Figure 4. The MEMS oscillator…

So, this paper requires a major revision.

Author Response

his submitted paper has 15 pages including 4 figure and 37 references. In their report, the authors suggest a simple frequency prediction for nonlinear oscillators with arbitrary initial conditions.

 Reply: Thanks for the positive comment.

The reviewer has looked through the paper and found that this presentation requires some revision.

First of all, the author must significantly improve the English language of their paper.

Reply: Thanks for the comment. English was checked carefully by each author and was finally confirmed by Prof. Khan.

Second, the References must be in the MDPI format.

Reply: Thanks for the suggestion. Done as requested.

Also, the Graphical Abstract should be deleted because this figure represents Figure 1a.

Reply: Thanks for the suggestion. Done as requested.

Figures 1, 2, 3, and 4 have 4, 4, 5, and 3 figures, respectively. They should be placed in two columns instead of the single column. Also, each figure of four should be situated on the same page.

 Reply: Thanks for the suggestion. Done as requested.

All the figure titles must start with “the”, for instance:

Figure 3. The comparison…

Figure 4. The MEMS oscillator…

 Reply: thanks for the suggestion. Done as requested.

So, this paper requires a major revision.

Round 2

Reviewer 1 Report

In the revised manuscript, the authors have made the requested corrections.

Reviewer 3 Report

A research article (revised manuscript ID: axioms-1454856-v2) entitled “A simple frequency formulation for the tangent oscillator” by a collaborative team from China and Saudi Arabia was submitted to the MDPI Axioms Journal.

This revised paper has 10 pages including 4 figure and 37 references. In their report, the authors suggest a simple frequency prediction for nonlinear oscillators with arbitrary initial conditions.

The reviewer has looked through the paper and found that this presentation requires some revision.

However, the author have significantly improved their paper.

The References must be in the MDPI format that can be done by a Journal staff. Ref. [33] has a chinese symbol that must be deleted.